# Deep Reinforcement Learning for Dynamic Capacitated Vehicle Routing Problem

## Abstract

Deep Reinforcement Learning (DRL) has become increasingly popular for solving Capacitated Vehicle Routing Problem (CVRP) due to its great potential. However, the current DRL models are only suitable for static environments where information about customers and orders is provided before the delivery vehicle departs from the depot and does not change during delivery. In reality, delivery tasks are dynamic, and much information about customers and orders is disclosed over time. In this paper, we propose a DRL model based on a designed dynamic attention network for dynamic CVRP, which extends the attention model from the original static-only CVRP environment to a dynamic CVRP environment. With dynamic encoder-decoder architecture, the proposed DRL model can track the changes in customer disclosure status in real-time. For comparison, we develop two methods based on LKH and OR-Tools for dynamic CVRP. Experimental results show that the DRL model outperforms LKH and OR-Tools in computational speed and solution quality. The code is publicly available on https://anonymous.4open.science/r/AM2DCVRP-0D4B.

## 1 Introduction

With the rapid growth of industries like e-commerce, food delivery, and ride-hailing, research on the Vehicle Routing Problem (VRP) is becoming increasingly relevant. The focus of research is shifting to address real-world environments, which include problems such as Capacitated VRP (CVRP) (Longo et al., 2006; Akhtar et al., 2017), Heterogeneous CVRP (HCVRP) (Koç et al., 2016; Subramanian et al., 2012; Li et al., 2021), and CVRP with Time Windows (CVRPTW) (Akbar et al., 2020; Baños et al., 2013). These problems are aimed at finding optimal routes under various constraints.

However, most of the research in the field of VRP is based on static delivery tasks. In such tasks, the information about customers and orders is provided before the delivery vehicle departs from the depot, and it remains constant during delivery. However, in real-world, the delivery tasks are often dynamic, where only some orders are known in advance before the vehicle departs from the depot and the rest of the orders are disclosed over time (Jia et al., 2017). Therefore, research in the field of dynamic VRP has begun to emerge in recent years.

Ulmer et al. (2019) devised a rollout algorithm based on offline value function approximation (VFA) that takes into account the temporal and spatial anticipation of service requests. This algorithm is designed to solve the single-vehicle routing problem with stochastic service requests. Sultana et al. (2021) developed a fast, parallelized, and approximate learning-based solution for the generic class of CVRP with Time Windows and Dynamic Routing (CVRP-TWDR). It allows each agent (vehicle) to independently evaluate serving each customer, and uses a centralized allocation heuristic to make the final allocations based on the generated values. Jia et al. (2017) created a dynamic logistics scheduling system using a set-based particle swarm optimization algorithm. Additionally, Meng et al. (2021) proposed a dynamic optimization policy that enhances Variable Neighborhood Search (VNS) to address CVRP, where transportation costs among the customers change over time. It takes into account both initial scheduling and rescheduling when traffic information changes. Liu (2019) introduced a mixed-integer programming (MIP) model, which discretizes the temporal dimension to exchange the spatial dimension's continuity. This allows for the dynamic input of order information with arbitrary pickup and delivery locations, thereby facilitating on-demand meal delivery service scheduling. These successful studies have paved the way for the development of dynamic VRP.

Besides previous traditional exact and heuristic algorithms, there has been a burgeoning interest in using DRL algorithms to solve VRP. Compared to other algorithms like exact and heuristic algorithms, end-to-end DRL algorithms can handle multiple tasks uniformly, offering faster computation and higher solution efficiency. Notable approaches include the attention model (AM) (Kool et al., 2018), multi-decoder attention model (Xin et al., 2021), and policy optimization with multiple optima (Kwon et al., 2020), all of which have shown promising results and have source code available. However, it is unfortunate that most DRL-based VRP research has focused on static cases. To the best of our knowledge, DRL for dynamic CVRP has not been well addressed yet. This paper proposes an approach for dynamic cases, providing inspiration for future DRL-based VRP research in dynamic environments.

When employing DRL algorithms to tackle the dynamic CVRP, there exist two key challenges that need to be addressed. The first challenge is about changes in customer information when new orders come in. The second challenge is related to the varying number of orders we get for different instances, which leads to different numbers of customer points. This can be a problem when we're doing batch training. To tackle the first challenge, we make changes to the input information whenever new orders are received. For the second challenge, we take inspiration from a method used in Wu et al.'s DRL-based improvement heuristic algorithms (Wu et al., 2021). This method involves adding a dummy depot to account for the varying lengths of solutions between instances. In our paper, we use the locations and demands of depot points to hide undisclosed customer points. This helps us address the issue of varying customer point numbers between instances, making batch training possible.

This paper presents three key contributions. First and foremost, we present a DRL model with a dynamic attention network and encoder-decoder architecture to address the dynamic CVRP, expanding it from a static environment to a dynamic one. Furthermore, we have developed two methods based on LKH and OR-Tools to serve as baselines in solving dynamic CVRP. Finally, we have established a mathematical model for dynamic CVRP, focusing on the constraints between the departure time at one point and the disclosure time of the succeeding point.

The rest of the paper is organized as follows. Section II introduces the mathematical model of dynamic CVRP. In Section III, the overall structure of DRL is first proposed, then followed by the specific implementation details, including the algorithmic pseudo-code. In Section IV, experimental results are presented to demonstrate the performance of the proposed DRL and LKH as well as OR-Tools method. Conclusions are given in Section V.

## 2 PROBLEM FORMULATION

### 2.1 PROBLEM DEFINITION

The dynamic CVRP delivery model involves a vehicle departing from a depot to deliver goods to multiple customer points, and the vehicle can return to the depot for restocking during delivery. Some customer points are known in advance, while others are gradually disclosed over time. To minimize unnecessary travel, the vehicle remains stationary after completing deliveries to all known customer points until the next point is disclosed. At the end of working hours, the vehicle finally returns to the depot. The vehicle's maximum capacity and the depot's location are already established. The objective is to plan routes to minimize the total distance traveled by the vehicle. To facilitate analysis and research, we have made the following assumptions: (1) The vehicle's speed remains constant at 1; (2) The demand for goods at each customer point does not exceed the vehicle's maximum capacity; (3) Each customer point is only serviced once.

### 2.2 MATHEMATICAL FORMULATION

Suppose there are $n$ customer points, and we have a set of locations $X = \{loc_0, \ldots, loc_i, \ldots, loc_n\}$, in which $loc_0$ represents the location of the central depot, and $loc_i (i \neq 0)$ represents the location of the $i^{\text{th}}$ customer point. We also have a demand set $D = \{d_1, \ldots, d_i, \ldots, d_n\}$, where $d_i$ represents the demand of the $i^{\text{th}}$ customer point, and a maximum material capacity $Q$ for the vehicle, such that $\forall i \in [1, n], d_i < Q$. Additionally, we have a set of disclosure times $T = \{t_1, \ldots, t_i, \ldots, t_n\}$, where $t_i$ denotes the time at which the $i^{\text{th}}$ customer point is disclosed. Some customer points are disclosed before the vehicle departs from the depot, and their disclosure time is 0. While visiting customer

point $j$, the set of all previously visited customer points is represented as $\pi_j$. The objective is to minimize the distance traveled by the vehicle while ensuring that each customer point's demand is met.

obj.

$$\min \sum_{i\in[0,n]} \sum_{j\in[0,n]} \mathrm{Eu\_2D}(i,j) \times r_{ij}, \tag{1}$$

s.t.

$$\sum_{j\in[0,n]} t_j \times r_{ij} \le t_i^{\mathrm{cur}}, \forall i \in [1,n] \cup (0,k), \tag{2}$$

$$t_i^{\mathrm{cur}} = \begin{cases} \min_l t_j, \text{if } \sum_{j\in[1,n]} t_j^{\mathrm{cur}} \times r_{ji} < t_l, \forall l \in ([1,n] - \sum_{j\in[1,n]} \pi_j \times r_{ji}) \\ 0, \text{else if } i = (0,1) \\ \sum_{j\in[1,n]\cup(0,k)} (t_j^{\mathrm{cur}} + \mathrm{Eu\_2D}(j,i)) \times r_{ji}, \text{else} \end{cases}, \tag{3}$$

$$\sum_{i\in[0,n]} r_{ij} = \sum_{i\in[0,n]} r_{ji} = 1, \forall i \in [0,n], \tag{4}$$

$$\sum_{i\in[1,n]} \sum_{j\in[0,n]} r_{ijk} d_i \le Q, \forall k. \tag{5}$$

In objective 1, the term $\mathrm{Eu\_2D}(i,j)$ represents the Euler distance between point $i$ and point $j$. Furthermore, $r_{ij}$ is a binary variable indicating whether a vehicle travels directly from customer point $i$ to customer point $j$. Another similar binary variable, $r_{ijk}$ in Equation 5, indicates if the vehicle departs from the depot during the $k$ departure. The information in constraint 2 implies that the vehicles do not visit any undisclosed customer points. According to the proposed model in this paper, the demands and locations of these undisclosed customer points are not known. The variable $t_i^{\mathrm{cur}}$ represents the time at which a vehicle departs from point $i$, while $t_{(0,k)}^{\mathrm{cur}}$ represents the time at which the vehicle departs from the depot for the $k^{\mathrm{th}}$ time. Within the specified constraint 3, we can observe the computation of the departure time $t_i^{\mathrm{cur}}$ from point $i$. If all disclosed customer points have already been visited, the vehicle will await the revelation of the next customer point. In such instances, the scope of values for $l$ encompasses all unvisited customer points. Conversely, if no unvisited customer points remain, the vehicle will depart straightaway upon reaching point $i$, foregoing any waiting period. Here, $i = (0,1)$ denotes the vehicle's first departure from the depot. As per constraint 4, the vehicle is allowed to enter and exit each customer point only once. Additionally, constraint 5 specifies that the total demand served by the vehicle during any departure from the depot until the next return to the depot should not exceed the maximum capacity of the vehicle.

## 3   AM FOR DYNAMIC CVRP

In the dynamic CVRP, the number of customer points may change over time as some of them are gradually disclosed. However, existing models employing the attention model have a fixed number of customer points during the solution and cannot accommodate dynamic changes. The algorithm proposed in this paper tracks customer point disclosure status by dynamically adjusting input information. The overall network structure, depicted in Figure 1, still comprises an encoder and a decoder. Nevertheless, the key innovation lies in the continuous adaptation of input information to reflect changes in the disclosure status. Encoding is initially performed using information for only $n_0$ ($n_l$ when step $l = 0$) customer points are disclosed before the vehicle's departure from the depot. Subsequently, every time a new customer point is disclosed, the input data is updated to incorporate information for all $n_l$ customer points that have been disclosed up to the $l^{\mathrm{th}}$ step in the loop.

The dynamic CVRP may involve a changing number of customer points as some are gradually disclosed. However, the current DRL models that utilize the attention model rely on a fixed number

of customer points throughout the solution and cannot adapt to these dynamic changes. To address this issue, this paper introduces a dynamic attention model that adjusts the input information in real-time to track customer point disclosure status. The network structure, as shown in Figure 1, employs a dynamic encoder-decoder architecture that continuously adapts to reflect changes in the disclosure status. Initially, the encoding process only considers information for $n_0$ (or $n_l$ at step $l = 0$) customer points that are disclosed before the vehicle departs from the depot. Subsequently, each time any new customer point is disclosed, its location and demand will add to the input data. So that, there are a total of $n_l$ customer points in the input data by the $l^{\text{th}}$ step of the loop.

The disclosure status is updated through a series of steps. Each step, labeled as $l$, calculates the next destination point of the vehicle as $i_{l+1}$ in Figure 1. The departure time $t_i^{\text{cur}}$ of the vehicle from point $i$ can be calculated using Formula 3. We compare $t_i^{\text{cur}}$ with the disclosure times of all points $T = \{t_1, \ldots, t_i, \ldots, t_n\}$. If the disclosure time is earlier than $t_i^{\text{cur}}$, the status is considered disclosed. Otherwise, it is considered undisclosed.

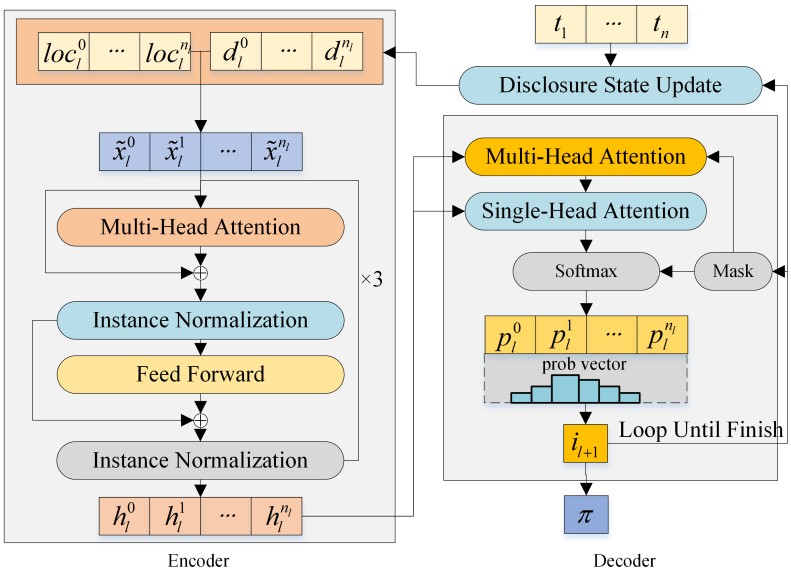

Figure 1: Architecture of our policy network.

In deep neural networks, batch training and batch inference play vital roles in achieving faster training, which is a significant advantage of such networks. However, customer points' disclosure times vary across different instances, which can cause certain instances to have disclosed customer points at a step, while others may not at the same step. As a result, the number of disclosed points, denoted as $n_l$, may differ at step $l$ in different instances, leading to varying input information lengths across the instances. This creates a challenge for both batch training and batch inference since they require consistent input information lengths for all instances.

To address this issue, we developed a method inspired by the DRL-based improvement heuristics algorithm proposed by Wu et al. (2021). Their approach involves improving an initial solution for each instance using DRL. However, the length of the path sequence for each instance's solution can vary due to the different numbers of actual returns to the depot. They added a sufficient number of dummy depots to the initial solution to address this issue. We build upon this method and further optimize the network as follows:

1. The input message length is set to a fixed value instead of a previously set variable value $n_l$ in the loop. In some real-world applications, the number of customer points disclosed during working hours may vary. Therefore, it is recommended to set the fixed value to the maximum number of customer points that may be encountered to ensure that all possible customer points are taken into account. For the purposes of this document, the value is $n$.

2. For undisclosed customer points, their locations and demands are replaced with those of the depot, and the demand of the depot is set to zero in this paper.

3. To ensure that no attention information of undisclosed customer points is calculated during decoding and their selection probability is zero, they are masked in both the Multi-Head Attention layer and Softmax process.

To better understand the proposed DRL algorithm in this paper, we present pseudo-code in Algorithm 1 that outlines the process of solving the dynamic CVPR. Initially (Line 1), sets the current time to zero, assumes a fully-loaded vehicle, and assumes that none of the customer points have been visited. At the start of the loop (Line 3), the disclosure status of each point is calculated by comparing the disclosure time of that point with the current time. Locations and demands of undisclosed points are transformed into those of the depot. Next (Line 4), the encoding information of the transformed points is processed. A mask matrix is computed, which includes undisclosed points, points with demands exceeding the remaining loaded material, and points that have already been visited. Simultaneously (Line 5), a mask matrix is obtained, where the information about undisclosed points, points with demands exceeding the remaining loaded material, and points that have already been visited is set to one, and that about other points is zero. Following this (Line 6), multi-head attention is not calculated for those points that are one in the mask matrix. Additionally (Line 8), the selection probability for these points is set to zero. Subsequently (Line 9), using a greedy method, the point with the highest probability $i_{l+1}$ is selected as the target point, which the vehicle travels directly to from the current point. Then (Line 10), the current time, vehicle load, and the visited status of customer points are updated. Eventually (Line 11), the planned route $\pi$ is updated by adding the target point $i_{l+1}$.

---

**Algorithm 1:** DRL for dynamic CVRP

---

**Data:** disclosure time set $T$, demand set $D$, location set of customer points and depot $X$, maximum capacity $Q$.
**Result:** $\pi$.

1  Initial $t^{cur} = 0, load = Q$, step $l = 0, visited$, net parameter $\theta$;
2  **while** *not finish all points* **do**
3      $D', X' = \text{Transform}(t^{cur}, T, D, X)$;
4      $encode = \text{ENCODE}_\theta(D', X')$;
5      $mask = \text{MASK}(T, t^{cur}, load, visited)$;
6      $mha = \text{MHA}_\theta(encode, mask)$;
7      $sha = \text{SHA}_\theta(mha, encode)$;
8      $p = \text{softmax}(sha, mask)$;
9      $i_{l+1} = \text{select}(p)$;
10     $t^{cur}, load, visited = \text{update}(i_{l+1})$;
11     $\pi = \pi + [i_{l+1}]$;
12     $l = l + 1$;
13 **end**
14 return $\pi$.

---

## 4 EXPERIMENTS

### 4.1 INDIRECT IMPLEMENTATION OF DYNAMIC CVRP

Currently, research on dynamic CVRP is limited, and there is a need for more open-source code to address this issue. Even widely used VPR-solving tools such as LKH and OR-Tools do not support dynamic CVRP solving, making it challenging to compare the proposed algorithm in this paper with other algorithms. To solve this problem, the paper presents a method to indirectly solve the dynamic CVRP through solving the static CVRP using LKH and OR-Tools. Although this method cannot completely replace the direct approach, it can still produce comparable results.

The method is illustrated in Figure 2. In Step 1, a path is planned as $[0, A, cur, B, 0, C, 0]$ for visiting all known customer points by solving the static CVRP. If a new point, such as the blue star point $new$ in Step 2, is disclosed while the vehicle is following the path just planned towards the red square point $cur$, the vehicle will no longer continue on its current path; instead, it will follow a newly re-planned route. In the diagram, the solid lines $[0, A, cur]$ represent the part already travelled of the path, whereas the dashed lines $[cur, B, 0, C, 0]$ depict the part yet to be taken.

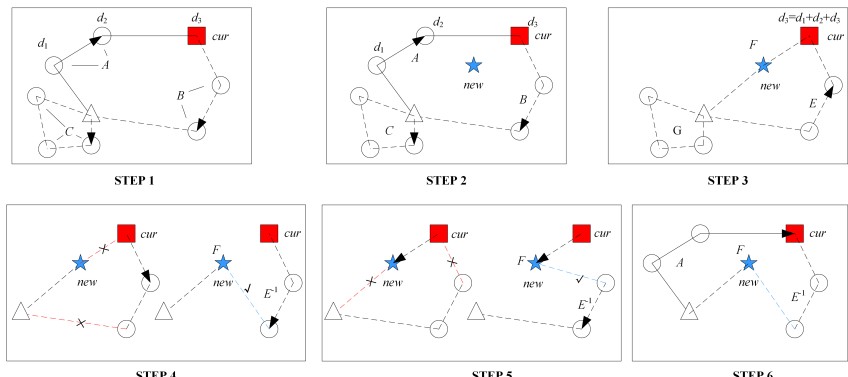

Figure 2: Indirect implementation of dynamic CVRP. $A, B, C, E, F, G$ are sequences of some points.

Several steps need to be taken to plan a new path that includes the newly disclosed point and remaining customer points. Firstly, exclude previously visited points, and set current demand to the sum of previously visited customer demands since the last departure from the depot. For instance, in Step 3, $d_3 = d_1 + d_2 + d_3$. After that, execute a static CVRP path planning for visiting the remaining points and the current point, i.e., $\{cur, new, B, C, 0\}$. Then, trim the path to make $cur$ the start point of the path. Suppose the newly planned path is $[0, E, cur, F, 0, G, 0]$. To trim the path with minimal modification, there are only two ways: reverse $[E, cur]$ and obtain $[0, cur, E^{-1}, F, 0]$; or reverse $[cur, F]$ and obtain $[0, E, F^{-1}, cur, 0]$. Then, delete the 0 near $cur$ to get $[cur, E^{-1}, F, 0]$ like in Step 4 and $[cur, F, E^{-1}, 0]$ like in Step 5, respectively, where $E^{-1}$ represents the reverse path of $E$. Since either method only deletes two links and adds one link, they change very little. Finally, combine the shorter one, which in this case is the former, with the previously visited path, and get $[0, A, cur, E^{-1}, F, 0, G, 0]$, as shown in Step 6. The vehicle follows this newly planned path until another customer point is disclosed, and then it re-plans the path using this method again until it eventually completes all customer points.

## 4.2 EXPERIMENTAL SETUP

This paper presents an implementation of the proposed algorithm using PyTorch and trains the policy network model on 10 GPUs (PH402 SKU 200 Tesla with 32GB of VRAM). The experiments were conducted on an Ubuntu 16.04.7 LTS operating system (GNU/Linux 4.4.0-31-generic x86-64). The case studies in this paper involve dynamic CVRP with customer point sizes of 10, 20, and 50. To evaluate performance, comparisons were made between the proposed algorithm and LKH as well as OR-Tools.

The parameter settings and data generation in this paper are identical to those of AM (Kool et al., 2018). Dynamic customer points account for 50% of the total, and disclosure times for tasks with sizes of 10, 20, and 50 are uniformly generated within the ranges [0-3.8], [0-6.4], and [0-10.98], respectively.

## 4.3 COMPARATIVE STUDY

Figure 3 displays the training curve for the objective value in the proposed algorithm, where "obj." represents the average transportation path length of all training instances (see Equation 1). From the training curve, it can be observed that the algorithm exhibits stable performance across different problem sizes, converging quickly, with smaller instances converging even faster.

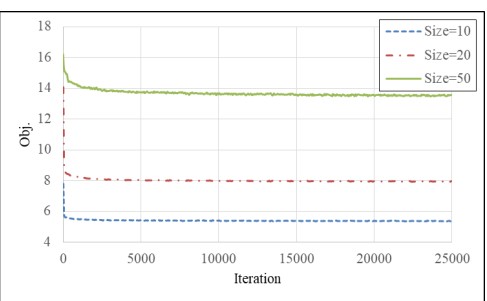

Figure 3: Training curve for obj.

After training the proposed algorithm, it was tested alongside LKH and OR-Tools. In the experiments, 1024 dynamic CVRP instances were randomly generated for each problem size, using a fixed random seed. The average length of the paths solved by each algorithm for these instances are presented in Table 1. The comparison revealed that the proposed algorithm consistently outperforms LKH. It shows slightly inferior performance to OR-Tools for smaller problem sizes (size=10 and 20), while exhibiting the best performance among the three for larger problem sizes (size=50). Additionally, the proposed algorithm significantly outperforms LKH and OR-Tools in terms of calculation time.

Table 1: Performance of each algorithm in 1024 dynamic CVRP instances.

| Size | LKH | | OR-Tools | | DRL | |
|---|---|---|---|---|---|---|
| | obj. | time | obj. | time | obj. | time |
| 10 | 5.88 | 1m8s | 5.40 | 1h7m25s | 5.48 | <1s |
| 20 | 8.71 | 9m3s | 8.12 | 2h23m58s | 8.14 | <1s |
| 50 | 14.88 | 1h45m8s | 14.11 | 6h7m43s | 13.97 | 21s |

During the experiment, OR-Tools faced several instances that it couldn't solve. For instance, in cases with a size of 50, almost 40% of the instances were unsolvable. To deal with these cases, this paper followed a skipping approach, which involved calculating the average for the remaining solvable 60% of instances as "obj.". In comparison, LKH uses a soft-constraint approach for unsolvable instances, adding a penalty value to the path length as a final path length for instances exceeding constraints.

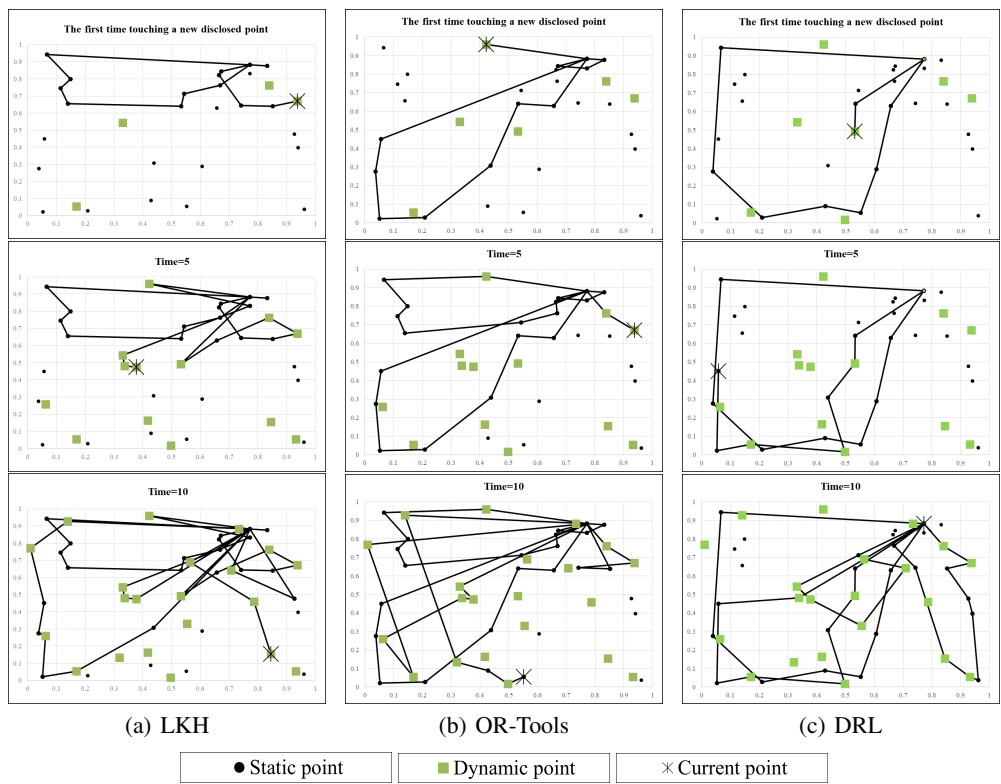

(a) LKH      (b) OR-Tools      (c) DRL

● Static point    ■ Dynamic point    ✳ Current point

Figure 4: Screenshots of the example solutions obtained by each algorithm at different times.

To better illustrate the differences between the algorithms, Figure 4 displays screenshots of example solutions obtained by each algorithm at different times. For a solution obtained by an algorithm, we captured three moments during the solving process, showing the disclosure status of customer points

and the vehicle's route. The first moment is when the vehicle first visits a newly disclosed point. The second moment is at 5, typically corresponding to the vehicle having visited approximately 1/3 of the customer points. The third moment is at 10, generally corresponding to the vehicle having visited about 2/3 of the customer points. In the figure, "static point" refers to customer points disclosed before the vehicle departs from the depot, "dynamic point" refers to customer points disclosed gradually after the vehicle departs from the depot until the current moment, and "current point" indicates the last location visited by the vehicle before the current moment. From the figure, it can be observed that there are no significant differences between the algorithms at the first moment. However, as time progresses, by the second moment, the path planned by LKH start to become disordered, with noticeable crossing between routes and longer path length. Although there are no significant differences between OR-Tools and our algorithm at this point, as time progresses to the third moment, the path planned by OR-Tools also start to become slightly disorderly compared to the latter.

## 5 CONCLUSION

This study focuses on addressing the dynamic Capacitated Vehicle Routing Problem (CVRP) by extending the previously static CVRP attention network to a dynamic environment. By utilizing a dynamic encoder-decoder architecture, the proposed Deep Reinforcement Learning (DRL) model is capable of adapting to changes in customer disclosure status. As there is currently no other paper that discusses DRL methods used to solve this CVRP where orders are disclosed over time, We believe that our method is a powerful starting point for DRL for dynamic CVRP.

To facilitate experimental comparisons, we have also devised two additional methods for solving dynamic CVRP using LKH and OR-Tools as baselines. Through extensive experimentation conducted on 1024 instances of varying customer sizes with a 50% dynamic ratio, DRL demonstrates superior performance compared to LKH and OR-Tools in dynamic CVRP.

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
