# OpenReview forum: "Deep Reinforcement Learning for Dynamic Capacitated Vehicle Routing Problem"
_ICLR.cc/2024/Conference — Submitted to ICLR 2024_

### Official Review · Reviewer_rVn7 · 2023-10-29

**Soundness:** 1 poor
**Presentation:** 2 fair
**Contribution:** 1 poor
**Rating:** 3
**Confidence:** 5

**Summary:**

This paper proposes a dynamic attention network for dynamic capacitated vehicle routing problem.

**Strengths:**

This paper proposes a dynamic attention network for dynamic capacitated vehicle routing problem.

**Weaknesses:**

This work seems to be not novel enough. The similar ideas of the proposed dynamic attention network can be found in previous literature [1]. The batch training for sequences with different length is a common issue in the field of natural language processing, and this issue can be easily tackled by padding.

The experiments are not sufficient and not convincing.

[1] Solving Dynamic Traveling Salesman Problems With Deep Reinforcement Learning, IEEE TNNLS 2023.

**Questions:**

See weaknesses.

---

### Official Review · Reviewer_9vhk · 2023-10-30

**Soundness:** 2 fair
**Presentation:** 2 fair
**Contribution:** 2 fair
**Rating:** 1
**Confidence:** 5

**Summary:**

This paper introduces a DRL-based framework for solving the dynamic CVRP, enabling the model to adapt to real-time customer location updates. The results achieved with the DRL-based model surpass those of the benchmark solutions. However, the presentation of the paper prevents readers from fully grasping the ideas and the implementation details. The mathematical formulation of the problem and the RL algorithm also needs significant work. This makes the validity of the outcomes and the reproductivity of the study questionable.

**Strengths:**

This paper offers innovative insights into the integration of DRL within the dynamic CVRP.
The paper demonstrates good performance compared to benchmark solutions.

**Weaknesses:**

I am concerned about the level of innovation presented in this paper. It appears that the only innovation is to set the input as dynamic disclosed customer locations in a mask fashion. Other than that, it follows a standard and straightforward pipeline that uses DRL to solve the static VRP problem, which has been well-studied in many other publications.

While I believe that DRL is used in this study, the paper provides very limited details / technical explanations related to DRL implementation and how DRL is specifically tailored to solve CVRP function, e.g., what are the rewards, what are the states, and how state transition is defined.

Moreover, the mathematical formulation in this paper is unclear and not rigorous.  For instance, the rationale behind the introduction of equations 1 to 5 is not clearly explained. Constraint 3 appears to define t_i^{cur} under various scenarios, but it requires further clarification.


Clarity, Quality, Novelty And Reproducibility

I had a hard time understanding the notations, often having to refer back to previous pages or struggling to find clear definitions. Examples include mha, sha, D', X', and π_{j} (is it a set or a sequence of visited nodes?).

Besides, the paper also contains multiple grammar issues, just to name a few:
Page 3-4: “However, the current DRL models that utilize the attention model rely on a fixed number of customer points throughout the solution and cannot adapt to these dynamic changes.”
Page 5: “The point with the highest probability il+1 is selected as the target point, which the vehicle travels directly to from the current point.“

Finally, I have concerns about the reproducibility of the paper due to the significant modifications made to transform the static VRP instance into a dynamic one.

**Questions:**

What is the motivation behind establishing the DCVRP formulation in this paper, rather than using the well-regarded models in the OR literature?

What are the fundamental/theoretical contributions of this paper that advance our understanding of Dynamic CVRP?

How do you formulate the DRL problem for the DCVRP problem? What are the drawbacks and challenges of using DRL in solving DCVRP,  and what are some problem-specific designs that contribute to tackling the challenges?

What are the parameter settings and scenario specifications for training and testing?

---

### Official Review · Reviewer_p28C · 2023-11-01

**Soundness:** 1 poor
**Presentation:** 1 poor
**Contribution:** 1 poor
**Rating:** 1
**Confidence:** 5

**Summary:**

This paper introduces a modified attention model tailored for the dynamic VRP, wherein orders are incrementally disclosed during the routing plan execution. The propsoed approach employs "dummy depots" as placeholders for yet-to-be-revealed positions, which are later substituted with specific newly revealed nodes (orders) to facilitate re-optimization of the route. Comparative results against LKH and OR-Tools are presented.

**Strengths:**

* The paper addresses the dynamic CVRP, a subject of great significance for real-world applications.
* The authors have released several GIFs in the code repo.

**Weaknesses:**

* The paper claims that "DRL for dynamic CVRP has not been well addressed yet." However, there seems to be a noticeable omission in the literature review. See the references below:
```
[1] Peng, Bo, Jiahai Wang, and Zizhen Zhang. "A deep reinforcement learning algorithm using dynamic attention model for vehicle routing problems." Artificial Intelligence Algorithms and Applications: 11th International Symposium, ISICA 2019, Guangzhou, China, November 16–17, 2019, Revised Selected Papers 11. Springer Singapore, 2020.
[2] Joe, Waldy, and Hoong Chuin Lau. "Deep reinforcement learning approach to solve dynamic vehicle routing problem with stochastic customers." Proceedings of the international conference on automated planning and scheduling. Vol. 30. 2020.
[3] Oren, Joel, et al. "SOLO: search online, learn offline for combinatorial optimization problems." Proceedings of the International Symposium on Combinatorial Search. Vol. 12. No. 1. 2021.
[4] Zhang, Zizhen, et al. "Solving dynamic traveling salesman problems with deep reinforcement learning." IEEE Transactions on Neural Networks and Learning Systems (2021).
[5] Li, Xijun, et al. "Learning to optimize industry-scale dynamic pickup and delivery problems." 2021 IEEE 37th International Conference on Data Engineering (ICDE). IEEE, 2021.
[6] Ma, Yi, et al. "A hierarchical reinforcement learning based optimization framework for large-scale dynamic pickup and delivery problems." Advances in Neural Information Processing Systems 34 (2021): 23609-23620.
[7] Basso, Rafael, et al. "Dynamic stochastic electric vehicle routing with safe reinforcement learning." Transportation research part E: logistics and transportation review 157 (2022): 102496.
[8] Pan, Weixu, and Shi Qiang Liu. "Deep reinforcement learning for the dynamic and uncertain vehicle routing problem." Applied Intelligence 53.1 (2023): 405-422.
```
* Given the existing literature, the contributions presented in this paper appear limited. The concept of "dummy depots" isn't novel, and the proposed re-optimization process could be perceived as potentially time-intensive. Furthermore, the adaptation of using AM for dynamic VRP seems relatively direct and incremental.
* The experimental setup might benefit from scaling, as the current research is conducted on a rather limited size (with only 50 nodes), which might not sufficiently demonstrate the model's robustness.
* The manuscript's style and structure seem more suited for a journal than a conference like ICLR. Moreover, there are areas in the paper that appear to be incomplete. For instance, some details and formulas seem to be missing or inadequately explained. Repetitive paragraphs are in Section 3 and there are various typos.
* The design of action space might need reconsideration. In the context of dynamic VRP, it could be worth exploring whether vehicles should be permitted to wait at certain locations, anticipating the release of subsequent orders. This might enable more strategic route planning, taking into account potential future orders.

**Questions:**

I encourage the authors to work further on this topic.

---

### Official Review · Reviewer_CZ1i · 2023-11-02

**Soundness:** 2 fair
**Presentation:** 2 fair
**Contribution:** 1 poor
**Rating:** 3
**Confidence:** 5

**Summary:**

A deep reinforcement learning approach for the dynamic capacitated vehicle routing problem (DCVRP) is proposed and evaluated on uniformly randomly generated instances. The main contributions of the approach are a "dynamic attention network" for addressing the DCVRP. They develop two methods to solve the DCVRP based on LKH and OR-Tools. The paper also proposes a mathematical model for the DCVRP.

**Strengths:**

The DCVRP has indeed seen less attention than offline problems, and considering dynamic problems with learned methods definitely presents an exciting research opportunity.

There is an argument to be made that attention networks must be adjusted to handle this type of problem. This paper does not make the contribution (1) in this respect so clear. What is new and what is old is very hard to determine; the overall method from Wu is not described.

**Weaknesses:**

I do not consider the last two contributions to be significant contributions for ICLR for two reasons. First, these are operations research (OR) contributions that ought to be evaluated by the OR community and not ICLR. But even if we accept them as thematically relevant, which they aren't, neither would be accepted in any OR venue in their current state.

Regarding contribution (2): the methods are not even close to the state-of-the-art. In the NeurIPS meets EURO competition the organizers provide a method for solving such dynamic problems (admittedly with time windows, but removing those would be trivial) based on Vidal's Hybrid Genetic Search (HGS), which is the state of the art. It's relatively simple to extend the offline problem in a naive way to support the dynamic version. The approaches offered here are a decade behind the current state of the art. Another option would be to use Ulmer's approach to dynamic routing problems. The author's cite his work, so I am not sure why it is then forgotten by the time we get to the experiments.

Regarding contribution (3): The mathematical model is not structured in a way that any OR journal would accept it without a serious revision. One could call it unconventional at best, i.e., the model is not a mixed-integer linear program (even though the problem is linear). There are good reasons for the standard formats used in the OR community, namely that they are easy to read and interpret, which this model is not.

The experiments are also rather weak: only randomly generated instances, and even the algorithm run times seem to be randomly chosen. Please see Accorsi et al. (2022) "Guidelines for the computational testing of machine learning approaches to vehicle routing problem". Furthermore, the extent of the experiments is very limited. There's no investigation of parameter sensitivity, ablation, etc. It is all missing.

**Questions:**

I have no questions.

---

### Official Review · Reviewer_zXXw · 2023-11-03

**Soundness:** 1 poor
**Presentation:** 1 poor
**Contribution:** 2 fair
**Rating:** 3
**Confidence:** 3

**Summary:**

This paper proposes a deep RL-based framework for dynamic capacitated vehicle routing problem. Because of the dynamic nature, the algorithm is able to serve customers even if they are added later. The paper then compared the proposed approach with the existing algorithms.

**Strengths:**

1. The main strengths seem to be that this work can adaptively address the dynamic customer addition problem, using deep RL algorithm it can route the vehicles in an adaptive manner.

2. The algorithm's performance is better compared to the existing algorithms.

**Weaknesses:**

1. The paper is not well-written to judge the main contributions. For example, deep RL-based approaches have been proposed [A1,A2]. The algorithm has not made any comparison compared to the other existing deep RL-based work.

2. The algorithm has not explained many things to get proper insights of the algorithm. We should not judge any algorithm's performance truly based on its performance rather how it is being developed (and the intuitions behind that). The reviewer did not get any intuition behind the algorithm. For example, the paper repeatedly mentioned about applying deep RL. What are the parameters for this deep RL problem? What is the reward? What are the transition probability models? How do those change dynamically?

3. The paper uses attention model, what is the logic behind using it.

4. The optimization problem in (1)--(5) is not clear. The notations have not been introduced before proposing the problem, hence, it is difficult to parse what it is doing. Besides, it contains another optimization problem as a constraint (in (2)), hence it is a bi-level optimization problem. How do the authors propose to solve this bi-linear optimization problem?


[A1]. Joe, Waldy, and Hoong Chuin Lau. "Deep reinforcement learning approach to solve dynamic vehicle routing problem with stochastic customers." In Proceedings of the international conference on automated planning and scheduling, vol. 30, pp. 394-402. 2020.

[A2]. Hildebrandt, Florentin D., Barrett W. Thomas, and Marlin W. Ulmer. "Opportunities for reinforcement learning in stochastic dynamic vehicle routing." Computers & operations research (2022): 106071.

**Questions:**

1. See the weaknesses.

---

### Meta-Review · Area_Chair_ijyj · 2023-11-22

**Metareview:**

This paper proposes a deep RL-based framework for dynamic capacitated vehicle routing problem. Because of the dynamic nature, the algorithm is able to serve customers even if they are added later. The paper then compared the proposed approach with the existing algorithms.

The key strengths of the paper include: This paper offers innovative insights into the integration of DRL within the dynamic CVRP. The paper demonstrates good performance compared to benchmark solutions.

The key weaknesses include:
1. The algorithm has not explained many things to get proper insights of the algorithm.
2. The optimization problem is not clear. The notations have not been introduced before proposing the problem, hence, it is difficult to parse what it is doing.
3. The mentioned contributions seem limited to multiple reviewers.
4. Experimental results could have been improved, including for scalability, parameter sensitivity, ablation studies.

**Justification For Why Not Higher Score:**

There is no response from the authors, and the reviews are consistent.

**Justification For Why Not Lower Score:**

N/A

---

### Decision · Program_Chairs · 2024-01-16

Reject